# α-Synuclein seeding activity in duodenum biopsies from Parkinson's disease patients

**Sarah Vascellari**[1][☯]*, **Christina D. Orrù**[2][☯], **Bradley R. Groveman**[2], **Sabiha Parveen**[2], **Giuseppe Fenu**[3], **Giada Pisano**[3], **Giuseppe Piga**[3], **Giulia Serra**[3], **Valentina Oppo**[3], **Daniela Murgia**[3], **Andrea Perra**[1], **Fabrizio Angius**[1], **Andrew G. Hughson**[2], **Cathryn L. Haigh**[2], **Aldo Manzin**[1], **Giovanni Cossu**[3‡], **Byron Caughey**[2‡]

**1** Department of Biomedical Sciences, University of Cagliari, Cagliari, Italy, **2** Laboratory of Neurological Infections and Immunity (LNII), Rocky Mountain Laboratories, National Institute of Allergy and Infectious Diseases (NIAID), National Institute of Health (NIH), Hamilton, Montana, United States, **3** S. C. Neurology and Stroke Unit, AOBrotzu, Cagliari, Italy

☯ These authors contributed equally to this work.
‡ These authors are joint senior authors on this work.
* svascellari@unica.it

**Data Availability Statement:** All relevant data are within the paper and its Supporting Information files.

## Abstract

Abnormal deposition of α-synuclein is a key feature and biomarker of Parkinson's disease. α-Synuclein aggregates can propagate themselves by a prion-like seeding-based mechanism within and between tissues and are hypothesized to move between the intestine and brain. α-Synuclein RT-QuIC seed amplification assays have detected Parkinson's-associated α-synuclein in multiple biospecimens including post-mortem colon samples. Here we show *intra vitam* detection of seeds in duodenum biopsies from 22/23 Parkinson's patients, but not in 6 healthy controls by RT-QuICR. In contrast, no tau seeding activity was detected in any of the biopsies. Our seed amplifications provide evidence that the upper intestine contains a form(s) of α-synuclein with self-propagating activity. The diagnostic sensitivity and specificity for PD in this biopsy panel were 95.7% and 100% respectively. End-point dilution analysis indicated up to $10^6$ $SD_{50}$ seeding units per mg of tissue with positivity in two contemporaneous biopsies from individual patients suggesting widespread distribution within the superior and descending parts of duodenum. Our detection of α-synuclein seeding activity in duodenum biopsies of Parkinson's disease patients suggests not only that such analyses may be useful in ante-mortem diagnosis, but also that the duodenum may be a source or a destination for pathological, self-propagating α-synuclein assemblies.

## Author summary

Misfolded α-Synuclein deposition is a hallmark of Parkinson's disease. The gastrointestinal tract may be an initial site of α-Synuclein aggregation, and its detection might be useful in the early diagnosis of Parkinson's disease. Here, we have used a rapid, ultrasensitive seed amplification assay (RT-QuICR) to show that pathologic α-Syn aggregates with prion-like self-propagating activity are in the upper intestine (duodenum) of Parkinson's disease patients. Our *intra vitam* detection of α-synuclein seeding activity in duodenum

**Funding:** This work was partially supported by Intramural Research Program of the National Institute for Allergy and Infectious Diseases, National Institutes of Health (BC). The funder had no role in study design, data collection and analysis, decision to publish, or preparation of the manuscript. CDO, BRG, SP, AGH, CLH and BC received a salary from Intramural Research Program of the National Institute for Allergy and Infectious Diseases, National Institutes of Health.

**Competing interests:** BC, CDO and AH are inventors on patent applications pertaining to aSyn RT-QuIC technology. The other authors have declared that no competing interests exist.

biopsies gave high diagnostic accuracy. Quantitation revealed high levels of seeds in duodenal tissue. Thus, our findings suggest that abnormal α-Synuclein seeds in the upper intestine might be both an early accurate biomarker for Parkinson's disease and cause of gut dysfunction.

## Introduction

Definitive diagnosis of Parkinson's disease (PD) relies on postmortem detection of disease-associated α-Synuclein aggregates (α-Syn$^D$) in the brain [1]. According to the "dual hit hypothesis", α-Syn$^D$ may start accumulating at the site of enteric nerves in the gut or the olfactory bulb, and progressively spread in stages along neural tracts to the brain [2]. Others have suggested a reverse path of α-Syn$^D$ spreading, which could originate from the brain and then reach the gastrointestinal (GI) tract [3]. Clinical and neuropathological evidence indicates that motor symptoms in PD are often preceded by GI dysfunctions [4, 5] leading some researchers to consider PD as a gut-brain disorder. One recent study showed α-Syn$^D$ in the colon of pre-symptomatic mice expressing human A53T α-Syn before detecting it in the brain [6]. It is also postulated that both the gut and brain first hypotheses are possible, which would lead to a new classification of PD, depending on the different initial site of α-Syn$^D$ accumulation. These include a "brain-first" subtype, in which pathology originates primarily in the brain, and a "body-first" subtype, characterized by premotor rapid eye movement behavior disorder (RBD) and GI impairment, in which pathological changes begin in the enteric or peripheral autonomic nervous systems, and follow a route that involves the brainstem [7–10].

These considerations make α-Syn$^D$ detection in intestinal tissues an attractive strategy for the early diagnosis of PD. However, conventional immunological-based techniques for detection of α-Syn$^D$ have produced conflicting results, likely due to the limited sensitivities of these assays [11]. One recent study suggested that immunohistochemistry (IHC) based methods are not suitable for the diagnosis of PD [12]. In contrast, an initial report by Emmi et al [13] showed IHC detection of α-Syn$^D$ in 100% of duodenum samples tested from a small number of PD patients. Other studies also propose that IHC analysis of layers deeper than mucosa and submucosa, such as muscularis propria or myenteric plexuses, for enteric phosphorylated α-Syn detection could be used to predict the onset of motor symptoms in PD [14]. Recently, ultrasensitive seed amplification assays (SAAs) have been applied to help address these controversial questions. Based in principle on the RT-QuIC assay platform developed for PrP prions [15, 16], the RT-QuIC-like SAAs for synucleinopathies are techniques that exploit the ability of α-Syn$^D$ to self-replicate similarly to prions, and to amplify trace amounts of pathological α-Syn in biological specimens (e.g., brain tissues, cerebrospinal fluid [CSF], submandibular glands, saliva, olfactory mucosa and skin) [17–25]. SAAs, such as α-Syn RT-QuIC [18, 19] and α-Syn PMCA [26], have been shown to detect α-Syn$^D$ in intestinal tissues. However, these studies detected α-Syn$^D$ seeds in only 55% of cases [27], or used postmortem samples [28]. In this study, we analyzed duodenum biopsies from PD and control patient cohorts with a relatively rapid α-Syn RT-QuIC assay (RT-QuICR), which uses a mutated version of the human α-synuclein protein and achieves faster and more specific detection of α-Syn$^D$ [29] than prior SAAs [18, 26]. Given reports of tau and α-Synuclein copathologies [30–34], we also checked a subset of the biopsies for tau seeding activity using a tau RT-QuIC assay [35]. We found α-Syn$^D$, but not tau, seeding activity in the intestinal mucosa (IM) tissue in 95.7% of PD cases, suggesting both diagnostic and potential pathological implications.

## Results

### Patient demographics and clinical information

Demographic and clinical information about the participants recruited for this study is given in Table 1. The mean age at the time of biopsy collection was 67 ± 8 years for PD cases and 58 ± 10 years old for healthy controls (HCs). The number of male and female patients within the PD and HC cohorts was similar. For PD patients, the mean disease duration at the time of intestinal biopsy collection was 14 ± 5 years. Thirteen of the 23 cases showed premotor RBD from 7 years before the onset of motor symptoms. At the time of biopsy collections all PD

**Table 1. Demographic characteristics of Parkinson's and healthy control patients.**

| ID | Diagnosis | Age[1] | Sex (M/F) | Disease duration[1] | Constipation score | premotor RBD[2] | UPDRS III score[1] | Duodenal biopsies tested by RT-QuICR |
|---|---|---|---|---|---|---|---|---|
|  | Parkinson's disease | 67±8 | 11/12 | 14±5 | 11±6 | 13 (57) | 25±7 | n = 30 |
| IM1 |  | 73 | F | 6 | 17 | yes | 23 | 2 |
| IM2 |  | 65 | M | 25 | 19 | yes | 19 | 1 |
| IM3 |  | 75 | M | 11 | 3 | no | 24 | 2 |
| IM4 |  | 65 | F | 15 | 14 | yes | 24 | 1 |
| IM5 |  | 65 | F | 10 | 16 | yes | 20 | 1 |
| IM6 |  | 76 | F | 14 | 19 | yes | 21 | 1 |
| IM7 |  | 58 | F | 11 | 10 | no | 20 | 1 |
| IM8 |  | 58 | M | 15 | 14 | no | 13 | 1 |
| IM9 |  | 66 | M | 17 | 18 | no | 28 | 1 |
| IM10 |  | 70 | M | 12 | 4 | yes | 38 | 1 |
| IM11 |  | 70 | F | 15 | 6 | yes | 23 | 1 |
| IM12 |  | 69 | F | 12 | 14 | yes | 27 | 1 |
| IM13 |  | 68 | M | 19 | 7 | no | 27 | 1 |
| IM14 |  | 74 | F | 14 | 5 | yes | 39 | 1 |
| IM15 |  | 57 | F | 20 | 6 | yes | 28 | 1 |
| IM16 |  | 69 | F | 12 | 21 | no | 14 | 1 |
| IM17 |  | 56 | M | 8 | 18 | no | 19 | 1 |
| IM18 |  | 77 | M | 6 | 15 | yes | 23 | 1 |
| IM22 |  | 56 | F | 19 | 2 | no | 38 | 2 |
| IM23 |  | 55 | M | 13 | 7 | yes | 26 | 2 |
| IM24 |  | 77 | M | 20 | 12 | no | 21 | 2 |
| IM26 |  | 58 | F | 8 | 4 | yes | 30 | 2 |
| IM27 |  | 74 | M | 13 | 7 | no | 23 | 2 |
|  | Healthy Controls | 58 ±10 | 3/3 |  |  |  |  | n = 12 |
| IM19 |  | 42 | M |  |  |  |  | 2 |
| IM20 |  | 59 | F |  |  |  |  | 2 |
| IM28 |  | 60 | M |  |  |  |  | 2 |
| IM29 |  | 72 | M |  |  |  |  | 2 |
| IM30 |  | 53 | F |  |  |  |  | 2 |
| IM31 |  | 62 | F |  |  |  |  | 2 |

[1] At time of biopsy (mean±SD)

[2] Number of cases with premotor RBD and related percentages in parenthesis: n (%)

Abbreviations: RBD, rapid eye movement behavior disorder; SD, Standard Deviation; UPDRS, Unified Parkinson Disease Rating Scale.

cases were on levodopa medication and the mean of Unified Parkinson's Disease Rating Scale (UPDRS) III motor score was 25 ± 7. Several PDs reported moderate/severe constipation as a common condition of GI impairment with an average score of 11 ± 6, ranging from a minimum score of 3 to a maximum of 21.

Upper GI biopsies were performed and the mucosa from proximal small intestine (duodenum) was taken from each subject and tested by RT-QuICR assay.

### Histology and immunohistochemistry of human intestinal mucosa biopsies

To visualize the general morphology and cellular composition of IM tissues, selected biopsies were examined after staining with hematoxylin and eosin (H&E). Representative samples from two patients showed that IM biopsies mainly contain the mucosa layer (Fig 1A), or mucosa associated to a thin layer of submucosa (Fig 1B). These findings suggest that standard endoscopic duodenal biopsies, although variable in size, always include the mucosa and may also include small amounts of the sub-mucosal layer. In both specimens, microscopic analysis did not reveal pathological findings. We also performed IHC for thyrosine hydroxylase (TH) and choline acetyltransferase (ChAT) but it was not informative due to the low sensitivity of the method (see S1 Fig and S1 Text).

### Assessment of IM tissue matrix inhibition of α-Syn$^D$ RT-QuICR assay

Previous studies had shown matrix inhibition of the RT-QuIC when testing tissues such as brain with higher concentrations of material [19, 36]. Therefore, to investigate the ability of the RT-QuICR to discriminate between PD and HC subjects, we initially tested a subset of IMs at $10^{-2}$ and $10^{-3}$ tissue dilutions (Fig 2). Our results showed inhibition of RT-QuICR amplification at $10^{-2}$, with most of the wells being negative for each sample. However, when seeding with a $10^{-3}$ IM a marked improvement was observed in the ability to discriminate between PD and HC samples. We therefore tested the remaining of the samples at $10^{-3}$ or further dilutions. Primary RT-QuICR fluorescence curves for representative PD (n = 3) and HC (n = 3) cases are shown in S2 Fig.

### Diagnostic sensitivity of RT-QuICR detection of α-Syn$^D$ in IM biopsies

We next performed a blinded RT-QuICR analysis of an additional panel of IMs. A total number of 42 duodenal biopsies from 23 PD cases and 6 HCs were tested. Overall, we observed positive RT-QuICR responses in 22 out of the 23 duodenal biopsies from PD patients, and negative responses for all 6 HCs (Fig 3). The sensitivity and specificity of the RT-QuICR were

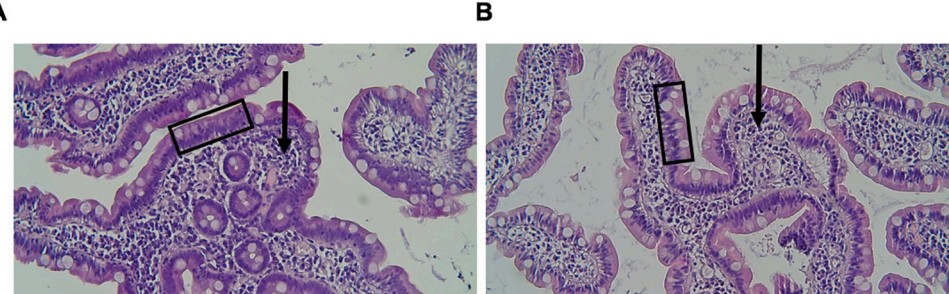

**Fig 1. Tissue morphology in duodenum intestinal mucosa biopsies. (A-B):** Representative photomicrographs of duodenum biopsies stained with hematoxylin and eosin. Both samples showed a normal morphology, well represented mucosa (black box) and variable amounts of submucosa (black arrow) depending on the biopsy specimen. Magnification x10.

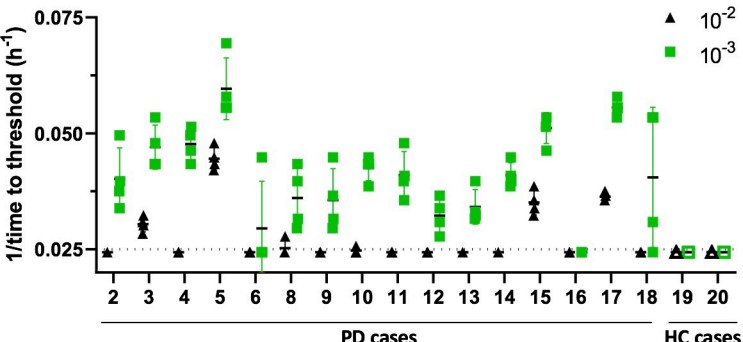

**Fig 2. Optimization of IM tissue dilutions used for RT-QuICR detection of α-Syn$^D$.** Inverse of the time to threshold for reactions seeded with IM tissue dilutions from PD or HC cases with the designated sample codes. Each symbol indicates a single reaction well with each sample analyzed in quadruplicate at both $10^{-2}$ (black triangles) and $10^{-3}$ (green squares) dilutions. A reaction time cut off of 40 h was used. The dotted line indicates the threshold for a positive reaction.

95.7% (95% CI: 78.1% to 99.9%) and 100% (95% CI: 54.1% to 100.0%), respectively (Table 2). Positive and negative predictive values (PPV and NPV) were 100.0% and 85.7% (95% CI: 46.9–97.6%), respectively. The value of diagnostic efficacy, expressed as the ratio between the sum of subjects classified as true positives and negatives, and the sum of the subjects classified as false positives and negatives, was 96.6% (95% CI: 82.2–99.9%).

## Quantitation of α-Syn$^D$ seeding activities in PD duodenum biopsies

To measure the levels of α-Syn$^D$ seeding activity in PD IM biopsies, end-point dilution analysis was performed [19]. IM samples were serially diluted from $10^{-3}$ to $10^{-6}$ for RT-QuICR analysis. Tissue concentrations of seeding units giving 50% positive replicate reactions (50% seeding doses or SD$_{50}$s) were then estimated using a modified Spearman-Kärber algorithm [37]. SD$_{50}$ concentrations were calculated for all the PD samples with the exception of samples 6 and 7, which did not have high enough seeding activity for the algorithm to be applied properly (Fig 4). On average, RT-QuICR-positive IMs from PD patients had 4.3 ± 0.8 log SD$_{50}$ per mg of tissue (range: 2.7 to 5.7).

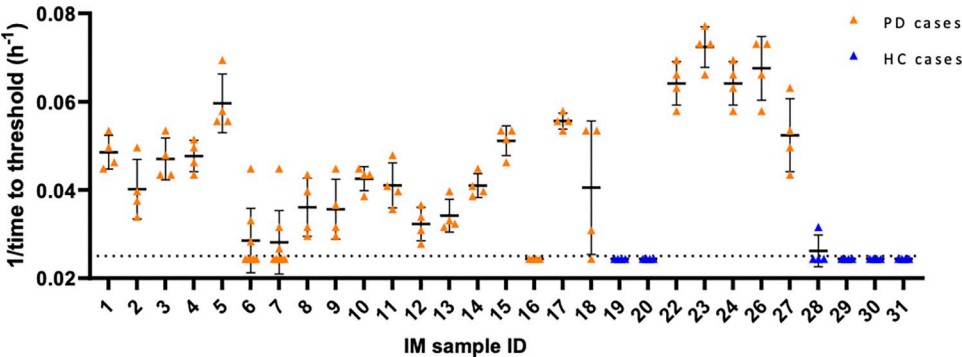

**Fig 3. RT-QuICR detection of α-SynD in duodenum IM biopsies.** Symbols represent the inverse of time to threshold for individual reaction wells seeded with a $10^{-3}$ IM tissue dilution from the designated PD or HC IM sample ID numbers. The dotted line indicates the threshold for a positive reaction. Bars show mean ±SD. Quadruplicate reaction wells are shown for each sample, with the exception of samples 6 and 7. The overall RT-QuICR status for the latter samples were inconclusive based on the initial quadruplicates. Upon repeating 6 and 7, as per the methods, both were determined to be positive overall and results from all 8 wells are shown here.

**Table 2. Diagnostic performance of IM α-Syn RT-QuICR.**

| Measures | Value | 95% CI |
|---|---|---|
| Sensitivity | 95.7% | 78.1% to 99.9% |
| Specificity | 100.0% | 54.07% to 100.0% |
| PLR | NA | NA |
| NLR | 0.04 | 0.01 to 0. 30 |
| Disease prevalence | 79.3% | 60.3% to 92.0% |
| PPV | 100.0% | |
| NPV | 85.7% | 46.9% to 97.6% |
| Youden's index | 0.95 | |
| Accuracy | 96.6% | 82.2% to 99.9% |

Abbreviations: PLR, Positive Likelihood Ratio; NLR, Negative Likelihood Ratio; PPV, Positive Predictive Value; NPV, Negative Predictive Value; CI, Confidence interval expressed as percentages; NA: Not available, the value cannot be calculated because the denominator is 0.

### RT-QuICR testing of two IM biopsies from individual patients

To address the question of α-Syn$^D$ distribution within the duodenum intestinal mucosa, we tested two samples collected during the same procedure from the superior and descending part of duodenum from 9 individual PD patients and 6 HC patients (Fig 5A). With the exception of patient 27, who was positive in only one of the two samples, RT-QuICR detected α-syn$^D$ in both biopsies from each patient. To evaluate whether α-Syn$^D$ levels were consistent across the duodenum sampling areas, end-point dilution analysis was performed on each specimen. Log SD$_{50}$s per mg of tissue (Fig 5B) varied by 0.5±0.4 between biopsies from the same patient.

### Lack of correlation between α-Syn$^D$ seed concentrations and clinical parameters

Spearman's correlation analysis was performed to investigate the relationship between the concentration of α-Syn$^D$ (log SD$_{50}$/mg) and clinical parameters. No significant correlation between SD$_{50}$ values and UPDRS III motor scores (r = − 0.23; p = 0.25; 95% CI:− 0.57% to 0.18; alpha = 0.05) (Fig 6A), constipation scores (r = 0.27; p = 0.18; 95% CI:− 0.14% to− 0.60; alpha = 0.05) (Fig 6B) and disease duration (r = 0.10; p = 0.64; 95% CI:− 0.31% to 0.47; alpha = 0.05) (Fig 6C) was observed.

### Lack of 3R or 3R+4R tau seeding activity in the IM biopsies

For comparative purposes, we also tested PD (n = 9) and HC (n = 1) IM biopsies for the presence of proteopathic seeds of another cytosolic protein, tau, using a tau RT-QuIC assay that ultrasensitively detects the tau seeds associated with either 3R (Pick's disease) or 3R+4R (Alzheimer's disease and chronic traumatic encephalopathy) tauopathies [35]. This comparison was of interest because of reported evidence of tau and synuclein co-pathologies in PD and Alzheimer's disease, and the detection of tau in Lewy bodies [30–34]. However, no tau seeding activity was detected in any of the IM specimens (Fig 7). As an indication that the IM tissue matrix did not prevent detection of tau seeds using this K12 tau RT-QuIC assay, we found that dilutions as extreme as $10^{-5}$ of sporadic Alzheimer's disease brain homogenates spiked into IM sample dilutions of $10^{-2}$ to $10^{-4}$ were readily detected. Simultaneous dilutions of the AD brain homogenates themselves showed positivity out to $10^{-8}$ dilutions, confirming the high

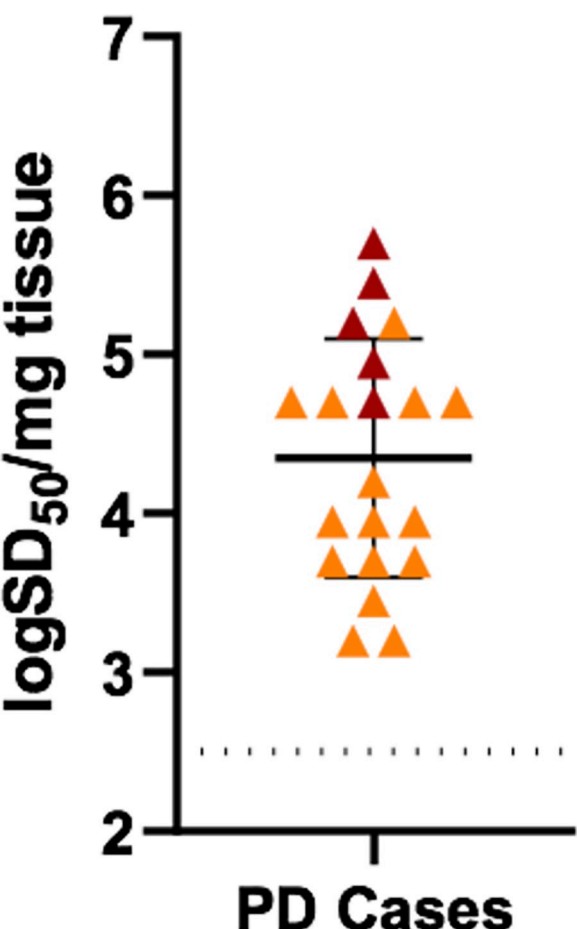

**Fig 4. RT-QuICR quantification of seeding activity in PD duodenum biopsies.** Orange triangles mark log SD$_{50}$/mg values from individual IM biopsies. Dark red triangles indicate values for PD IMs that did not completely reach end-point (0/4 positive wells) in our serial dilutions and therefore may be slight underestimations of the seeding activity by Spearman-Kärber analysis. However, each of those cases diluted to 2/4 or 1/4 positive wells which is at, or below, the definition of 1 SD$_{50}$, respectively, in those diluted aliquots, meaning that values shown in the figure should be close to being accurate. The dotted line indicates the RT-QuICR detection limit.

sensitivity of the assay for AD tau seeds. These tau RT-QuIC assay results provided evidence that no measurable 3R or 3R+4R (Alzheimer's-like) tau seeding activity accompanied the duodenal α-Syn$^D$ seeds of PD.

## Discussion

In this study, we have detected α-Syn$^D$, but not tau, seeding activity in enteric duodenal biopsies from patients with a clinical diagnosis of Parkinson's disease. We estimated the average α-Syn$^D$ seeding activity to be 4.3 log SD$_{50}$ per mg of tissue, which was similar to that reported for RT-QuIC analysis of both post-mortem colon and skin specimens from PD patients [20, 28]. This means that the IM seeding activity was comparable to levels that we have seen in various PD brain specimens, and >3 logs higher than levels typically seen in antemortem PD CSF samples [19, 25]

The diagnostic accuracy of RT-QuICR for PD in our initial duodenum biopsy panel was high, i.e., 95.7% sensitivity and 100% specificity. By comparison, a previous analysis of lower intestinal biopsies from a small cohort of PD and healthy subjects using an alternative SAA

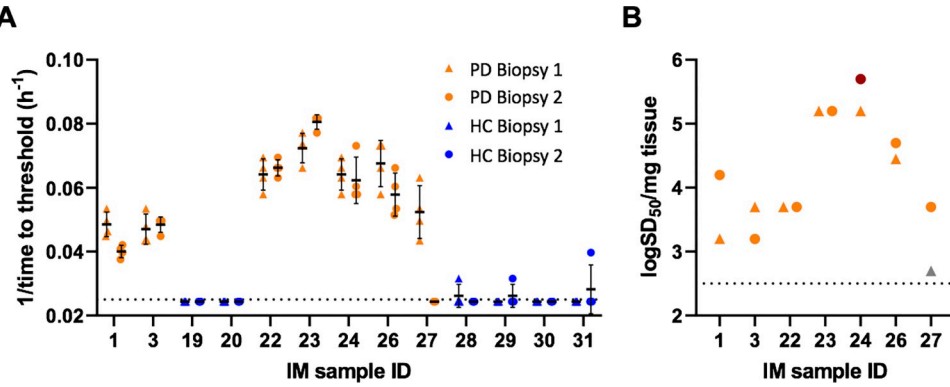

**Fig 5. Comparison of seeding activities in IM samples collected from first and second duodenum segments from individual patients. (A)** Inverse of time to fluorescence threshold comparisons for PD (orange) and HC (blue) cases. Each IM was analyzed in quadruplicate at $10^{-3}$ dilution, except for sample 27 which shows values from 8 reaction wells, with each symbol representing an individual reaction well from either the first (biopsy 1; triangles) or second (biopsy 2; circles) duodenum segments from the given patient. **(B)** Log $SD_{50}$/mg comparisons from end-point dilution analysis. The dark red circle represents a PD IM that did not reach end-point, and thus is likely an underestimate of the actual value. Grey triangle indicates a positive sample for which a log $SD_{50}$/mg value could not be accurately calculated due to low seeding activity. Log $SD_{50}$/mg for the first samplings are the same as those reported in Fig 2. The dotted lines indicate the threshold for detection (A) and the assay's detection limit (B).

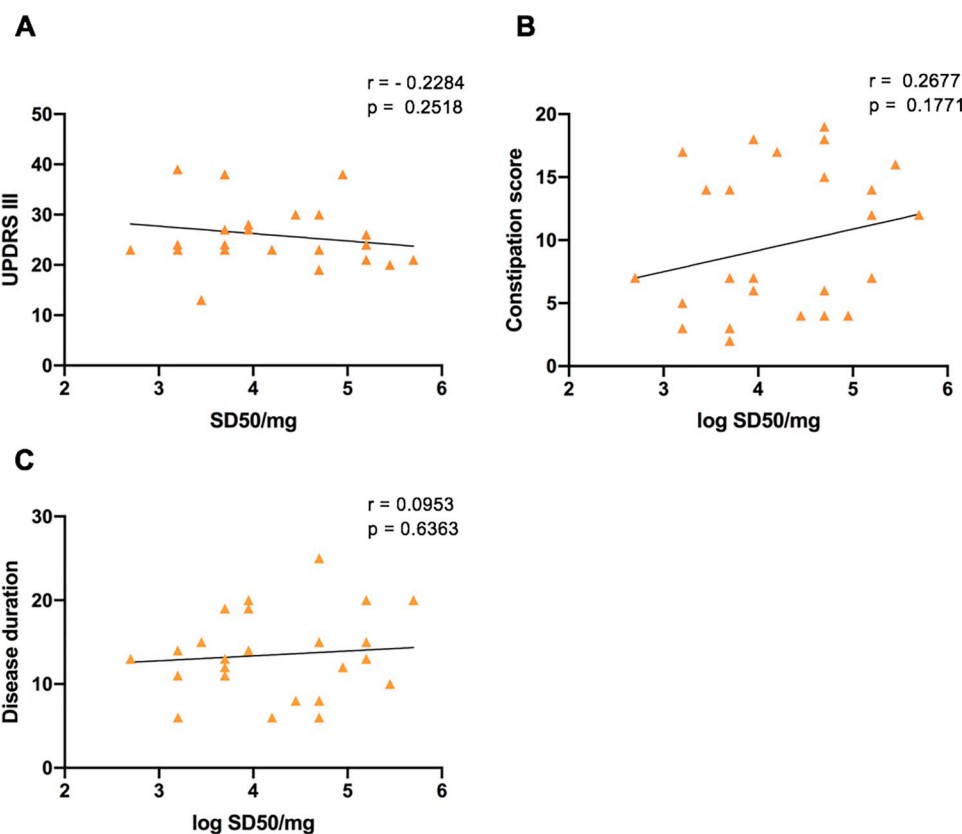

**Fig 6. Spearman correlation between α-syn$^D$ seed concentration (logSD50/mg tissue) and clinical parameters. (A)** UPDRSIII score versus IM log SD50/mg while patient was on Levodopa medication. **(B)** Constipation score and **(C)** Disease duration versus seed concentration. Spearman's r and p values are inset.

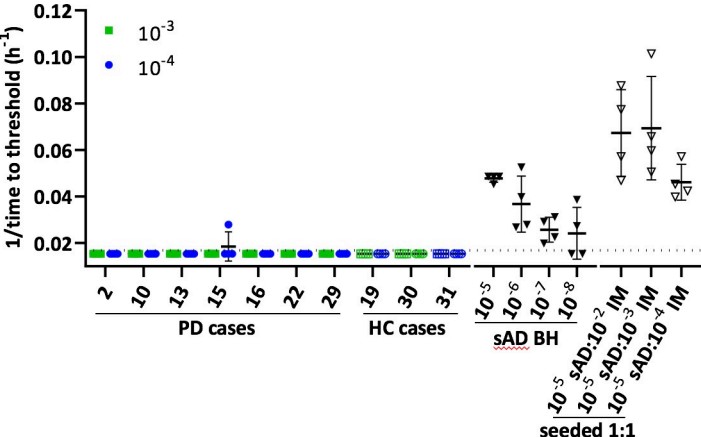

**Fig 7. Tau RT-QuIC analysis of PD IM specimens.** Colored symbols represent the inverse of the time to threshold for individual reaction wells seeded with a $10^{-3}$ (green squares) or $10^{-4}$ (blue circles) IM tissue dilution from the designated PD or HC cases. Solid downward triangles indicate control reactions seeded with sAD brain homogenate (BH) of the designated dilution with respect to solid tissue. Open downward triangles display representative data from reaction wells seeded with 1:1 mixtures $10^{-5}$ sAD BH and the designated tissue dilution of HC IM30. The dotted line indicates the threshold for a positive reaction. Error bars show mean ±SD. Quadruplicate reaction wells are shown for each sample.

previously detected α-Syn$^D$ with a diagnostic sensitivity of 55.6% and a specificity of 90.9% [27]. The reason for the increased sensitivity and specificity that we observed in the duodenum biopsies is unknown, but it could be due to the differences in the section of the intestine tested (rectum vs. duodenum) in the two patient cohorts and/or to the version of SAA used [19, 26]. α-Syn$^D$ can spread from gut-to-brain and brain-to-gut via vagal pathways in animal models of body-first and brain-first PD (reviewed in [10]). Since vagal innervation is highest in the upper gastrointestinal tract, one might expect to detect more α-Syn$^D$ in duodenal biopsies, compared to rectal biopsies. Emmi et al. recently detected α-Syn aggregates in duodenum biopsies from 22 patients with PD using IHC [13]. Our findings are consistent with these IHC results, and show additionally that duodenal α-Syn$^D$ has self-propagating (seeding) activity. We found similar amounts of seeding activity in contemporaneous biopsies from superior and descending sections of duodenum from individual patients, suggestive of widespread α-Syn$^D$ distribution within these duodenum segments. However, we assume that multiple IM collections, when possible, would improve diagnostic sensitivity. Emmi et al. [12] studied the relationship between the extent of α-Syn$^D$ and the severity of extrapyramidal symptoms, but failed to find a correlation among α-Syn$^D$ immunoreactive areas and the severity of motor symptoms, the global cognitive scales' score, the quality of life, and the UPDRS scale values. Likewise, we found no significant correlation between the α-Syn$^D$ seed concentrations and UPDRS III motor scores in patients on levodopa treatment. Thus, the link between the distribution of α-Syn$^D$ in peripheral tissues and the stage of disease remains unclear [38]. The extent to which factors such as bilirubin content and gut inflammation might affect α-Syn$^D$ seed concentrations and confound the interpretation of IM RT-QuICR results remains to be determined.

Although the high diagnostic sensitivity and specificity were observed in this study, our results cannot be validated by autopsy as the patients were still alive at the time this study was completed. However, the study participants were carefully chosen based on clinical criteria. Additionally, the number of patients was limited due to the availability of patients undergoing PEG-J tube placement, patient hesitation to undergo an additional, albeit minimally invasive procedure, and restrictions placed on elective medical procedures during the Covid-19

pandemic. We caution that PEG-J for continuous L-DOPA infusion is generally used in more advanced cases of PD where oral L-DOPA is less consistently effective. Thus, the RT-QuICR results that we have obtained with our cohort of more severe PD cases may not be generalizable to patients with more moderate/mild PD that can be controlled well via oral L-DOPA. Another caveat is that the average age of the PD cases was 9 years older than the non-PD controls, and, at this time we cannot exclude the possibility that this age differential contributed in some way to our divergent RT-QuICR results for these two cohorts.

In our study, 1 of 23 PD cases was RT-QuICR negative (IM16). At the time of IM homogenization, we noted that the tissue of this patient had an unusual hardened, thickened, and yellow appearance. A similar observation was made in one duplicate sample from IM27, which also gave an RT-QuICR-negative result. Because of this appearance, we speculate that these specimens might have been contaminated with bilirubin derived from the breakdown of hemoglobin in senescent red blood cells. Because blood contamination can inhibit some RT-QuIC assays [39], it is possible that residual blood components may have interfered. We also observed that in one PD patient (IM7), the RT-QuICR reaction gave a weak positive signal. This sample was collected from a patient affected by scleroderma, a condition which is known to damage peripheral nerves and mucosa [40] in the GI tract, most frequently in the duodenum [41]. Therefore, we hypothesize that the weak positive result for this subject could be due to vascular ischemia and fibrosis associated to scleroderma, which may have altered α-Syn$^D$ deposition and distribution.

Recent histopathological evidence indicates that enteric α-Syn$^D$ deposition is more prevalent in RBD-positive than RBD-negative subjects [42]. Moreover, subjects with premotor RBD and constipation fit the criteria of "body first" phenotype [8, 43]. In our cohort about half of the patients (52.2%) presented both premotor RBD and GI symptoms of constipation. Interestingly, the RT-QuICR-negative PD patient was also RBD-negative. However, other than this case, no significant differences in RT-QuICR detection or parameters were observed between RBD positive or negative subjects.

In conclusion, RT-QuICR provides a method for testing duodenal biopsies with similar sensitivity and specificity to SAAs of other diagnostically relevant samples [18, 19, 22, 25, 29] and skin [20, 23]. Paired with end-point dilution analysis, this assay can quantitatively assess levels of seeding activity among patients as well as across tissues, or in different sections of the same tissue. This could prove to be valuable for monitoring disease progression or selecting cohorts for clinical trials and monitoring their progress. While further studies are needed to elucidate the feasibility and utility of IM RT-QuICR, the potential for *intra vitam* monitoring and diagnosis, for example during routine esophagogastroduodenoscopy and colonoscopy screenings, holds promise for the early diagnosis of PD.

## Materials and methods

### Ethics statement

This study was approved by the Institutional Ethics Committee (Prot.PG/2017/17817) of the Azienda Ospedaliera Universitaria di Cagliari, Italy. All participants signed written informed consent documents.

### Participants

This study was approved by the Institutional Ethics Committee (Prot.PG/2017/17817) of the Azienda Ospedaliera Universitaria di Cagliari, Italy. All participants signed written informed consent documents. A total of 29 participants in this study, 23 PD patients and 6 non-neurodegenerative healthy controls (HCs) subjects, were recruited between September 2020 and

September 2022 at the Neurology and Digestive Endoscopy Units of AO Brotzu Cagliari. Idiopathic PD patients were diagnosed according to the UK Brain Bank criteria, and patients with atypical parkinsonism were excluded. All PD patients were evaluated by the Movement Disorder Society UPDRS III and IV and by the Non-Motor Symptom Scale (NMSS). The presence of premotor RBD was assessed retrospectively through the REM sleep behavior disorder screening questionnaire (RBDSQ) [44].

The criteria for evaluation of constipation was based on the Constipation Scoring System [45]. A total score ranging from 0 (no constipation) to 30 with higher scores indicating more severe constipation was assigned retrospectively to each patient in according to eight variables [45], such as frequency of bowel movements, painful evacuation, incomplete evacuation, abdominal pain, length of time per attempt, assistance for defecation, unsuccessful attempts for evacuation per 24 hours, and duration of constipation. Exclusion criteria were intestinal cancer, the concomitant presence of neurological, or unstable psychiatric illness or cognitive impairment. All PD subjects included in this study were given levodopa-carbidopa intestinal gel (LCIG). HCs were matched to PD subjects for sex and age and had no neurological or cognitive disorders.

## Intestinal mucosa collection

All PD patients underwent upper GI endoscopy for placement of an administration jejunal extension tube (PEG-J) for continuous levodopa enteral infusion, and intestinal mucosa biopsy. HCs were selected from those undergoing upper GI endoscopy biopsy for diagnostic investigations and surveillance.

Two to four IM biopsy specimens from the proximal small intestine (duodenum) were taken for each participant (~20 mg of total tissue). All samples were collected in sterile containers containing 20 to 50 mL of physiological solution and immediately frozen at –80˚C until analysis. Two additional IM biopsies were collected in sterile containers with 10% formalin for histological analysis and immunohistochemistry (IHC) examination.

## Histology and immunohistochemical analysis

Histological examination was done on hematoxylin and eosin stained sections. Four μm thick sections were cut from formalin fixed-paraffin embedded biopsies. After rehydration, sections were incubated for 20 minutes with ready to use Carazzi's Haematoxylin (Bio-Optica, Milan), soaked in tap water and counter-stained with 1% aqueous eosin Y solution for 20 seconds (Bio-Optica). Tissue examination was performed using a Leica DM6000 light microscope.

## Homogenization of intestinal mucosa samples

Intestinal mucosa samples were homogenized following the protocol described by Bargar et al [19, 28] with some modifications. In brief, IMs were thawed and washed three times in 1× phosphate buffered saline (PBS) to remove any residual blood. Samples were weighed and homogenized at 10% (w/vol) in ice-cold homogenization buffer containing 1 x PBS, 0.1% Triton X-100, 150 mM sodium chloride (NaCl), 5 mM of ethylene-diamino-tetraacetic acid (EDTA) and Complete Protease Inhibitor without EDTA (Roche). IM homogenization was done using zirconia beads (1 mm) in a mini-Beadbeater-16 device (Bead Mill$_{24}$; Thermo Fisher) at maximum speed. Samples were homogenized with 4 rounds of 1 min each at max speed, with cooling in between rounds. Next, samples were centrifuged at 500xg for 5 min and the supernatant was distributed in single use aliquots to be stored at -80˚C for subsequent RT-QuICR analysis.

## Recombinant α-Syn (K23Q) and tau (K12CFh) protein purification

Human K23Q rec α-Synuclein was purified as described by Groveman et al [19]. In brief, BL21(DE3) *Escherichia coli* bacteria were freshly transformed with the plasmid for recombinant K23Q α-Syn protein expression. One liter cultures were inoculated with a single colony and grown overnight using auto-induction media [46] containing 50 μg/mL kanamycin. The cells were lysed and the periplasmic fraction was isolated using an osmotic shock and acid purification protocol modified from Paslawski et al. [47]. The protein extract was filtered and loaded for purification into a 5 ml Ni-NTA column (Cytiva) on an Äkta Start chromatography system (GE). Peak fractions where then further purified using a 5 ml Q-HP column (Cytiva). The purified protein was filtered, diluted to 1 mg/ml, and dialyzed overnight at 4˚C using a 3 kDa MWCO dialysis membrane. Protein concentration was determined with a UV–VIS spectrophotometer using a theoretical extinction coefficient at 280 nm of 0.36 (mg/mL) − 1 cm − 1. Lyophilized protein aliquots were stored at − 80˚C. Human K12CFh rec tau substrate was purified as described by Metrick et al [35].

## IM α-Syn RT-QuICR and tau RT-QuIC assays

For α-Syn RT-QuICR assays 10% IM homogenates were thawed at room temperature, serially diluted in PBS containing N2 media supplement (Gibco) and tested by RT-QuICR using purified recombinant K23Q α-Syn as substrate. In brief, RT-QuIC buffer composition was: 40 mM phosphate buffer pH 8, 170 mM NaCl, 0.1 mg/ml K23Q rec-αSyn, 10 μM Thioflavin T (ThT). A volume of 98 uL of RT-QuIC buffer was loaded into wells of a black 96-wells plate with a clear bottom (Nunc). Plates were preloaded with six silica beads (0.8 mm, molecular biology grade; OPS Diagnostics) and reactions were seeded with 2 μL of IM homogenates at the specified dilutions for a final reaction volume of 100 μL. Next, plates were sealed (Nunc International sealer) and incubated in a BMG Fluostar plate reader at 42˚C for 60 h with cycles of 1 min shake (400 rpm double orbital) and 1 min rest. ThT fluorescence measurements (450 +/− 10 nm excitation and 480+/− 10 nm emission; bottom read) were taken every 45 min. All IMs were tested with at least 4 replicate reactions per sample dilution. Single reactions were considered positive when their fluorescence exceeded our threshold of 10% of the maximum value in the plate. IM samples were considered positive when ≥50% of replicate wells were above our ThT threshold within the established 40-h time cutoff. Samples that had 1 out of the 4 replicate reactions score as positive were deemed inconclusive and their analysis was repeated. Then, if the total percentage of positive reaction wells exceeded 25% then the sample was considered positive.

The K12 tau-RT-QuIC assay was performed as described by Metrick et al [35]. Briefly, the reaction buffer composition was: 40 mM HEPES, pH 7.4, 400 mM NaF, 40 μM heparin, 10 μM ThT, 0.1 mg/mL K12 CFh substrate. A volume of 48 μL of reaction buffer was added into wells of a black 384-wells plate and seeded with 2 μL PD IM homogenate dilutions alone or spiked with $10^{-5}$ AD brain at 1:1 volumetric ratios with the designated dilution of PD IM. Plates were sealed with sealing tape and placed in an Omega FLUOStar plate reader at 42˚C and subjected to rounds of 1 min shaking, 500 rpm, orbital, and 1 min rest, with ThT fluorescence reads (450 excitation, 480 emission) taken every 15 min. Reaction wells were considered positive on an individual reaction basis when their fluorescence value exceeded a threshold value of the mean fluorescence value plus 100 times the standard deviation from the first 5 h of readings.

## End-point dilution analysis and 50% seeding dose (SD$_{50}$) calculation

IMs were serially diluted 10-fold for end point analysis. Two μl of each IM dilution ($10^{-3}$ to $10^{-6}$) were used to seed individual reactions. RT-QuICR testing was done as reported above.

The amount of sample giving 50% positive replicate reactions ($SD_{50}$) was estimated as described by Srivastava et al. [37].

## Statistical analysis

Student's test and Fisher's exact test was used to analyze differences in demographic characteristics between cases and controls. Variability of the data was presented as mean ± standard deviation or as percentages. Measures of diagnostic performance, such as sensitivity, specificity, negative predictive value, and positive predictive value, were determined for comparisons between clinical diagnosis and RT-QuICR results. Measures were expressed as percentages with 95% confidence interval (95% CI). Spearman's rank correlation coefficient was used to determinate the correlation between SD50 values and motor UPDRSIII scores, disease duration and constipation score (95% CI; alpha = 0.05). All statistical analyses were performed and plotted using the Prism software (v.8 GraphPad). Statistical significance was set at $p < 0.05$.

## Supporting information

**S1 Fig. Immunohistochemistry analysis of human duodenum IM biopsies. (A)** Immunohistochemistry for tyrosine hydroxylase (TH). **(B)** Immunohistochemistry for choline acetyltransferase (ChAT).
(TIF)

**S2 Fig. Representative ThT fluorescence curves in IM α-Syn RT-QuICR. (A)** Traces from single reactions (n = 4 per case) seeded with $10^{-3}$ dilutions of duodenum IM biopsies from PDs cases 4, 5, and 15 (as colored) as a function of reaction time. **(B)** Traces from healthy control duodenum IM biopsies from cases 29, 30, and 31.
(TIF)

**S1 Data. Fig 2 tab**: Inverse time to threshold values ($h^{-1}$) for individual quadruplicate wells seeded with $10^{-2}$ or $10^{-3}$ dilutions of IM samples with the IM sample ID numbers shown in the "Patient" column. **Fig 3 tab**: Inverse time to threshold values ($h^{-1}$) for individual quadruplicate wells seeded with $10^{-3}$ dilutions of IM samples with the IM sample ID numbers shown in the "Patient" column. For samples 6 and 7, data from an additional 4 wells are shown. **Fig 4 tab**: log $SD_{50}$/mg tissue values calculated for individual IM biopsies from end-point dilution data. **Fig 5A tab**: Inverse time to fluorescence threshold comparisons between first (biopsy 1) or second (biopsy 2) duodenum segments from a given patient for the IM sample ID numbers shown in the "Patient" column. Data are from quadruplicate wells seeded with $10^{-3}$ IM dilutions. **Fig 5B tab**: Log $SD_{50}$/mg tissue values calculated for two IM biopsies from the same patient from end-point dilution analysis. **Fig 6 tab**: Individual IM log $SD_{50}$/mg tissue values, UPDRSIII scores, constipation scores, and disease durations (y) associated with the designated IM sample ID in the "Patient" column. **Fig 7 tab**: Top 2 rows: Inverse time to threshold values ($h^{-1}$) for individual quadruplicate wells seeded with $10^{-3}$ or $10^{-4}$ dilutions of IM samples with the IM sample ID numbers shown in the "Patient" row. sAD group: Analogous data from reactions seeded with $10^{-5}$ or $10^{-8}$ dilutions of sAD brain homogenate. The sAD spiked $10^{-5}$ group: data from reaction wells seeded with 1:1 mixtures $10^{-5}$ sAD BH and the designated tissue dilution of HC IM30. **S2A and S2B Figs tabs**: Quadruplicate ThT fluorescence readings as function of reaction time (h) for reactions seeded with $10^{-3}$ dilutions of IM sample IDs given the top row.
(XLSX)

**S1 Text. Immunohistochemistry analysis of human intestinal mucosa biopsies.**
(DOCX)

## Acknowledgments

The authors kindly acknowledge the Digestive Endoscopy Unit of AO Brotzu Cagliari for the assistance with subject recruitment.

## Author Contributions

**Conceptualization:** Sarah Vascellari, Christina D. Orrù, Giovanni Cossu, Byron Caughey.

**Data curation:** Sarah Vascellari, Christina D. Orrù, Bradley R. Groveman, Sabiha Parveen, Giuseppe Fenu, Giada Pisano, Giuseppe Piga, Giulia Serra, Valentina Oppo, Daniela Murgia, Andrew G. Hughson, Giovanni Cossu.

**Formal analysis:** Sarah Vascellari, Christina D. Orrù, Bradley R. Groveman, Sabiha Parveen, Andrew G. Hughson.

**Funding acquisition:** Byron Caughey.

**Investigation:** Sarah Vascellari, Christina D. Orrù, Bradley R. Groveman, Sabiha Parveen, Giuseppe Fenu, Giada Pisano, Giuseppe Piga, Giulia Serra, Valentina Oppo, Daniela Murgia, Andrea Perra, Andrew G. Hughson, Giovanni Cossu.

**Methodology:** Sarah Vascellari, Christina D. Orrù.

**Project administration:** Sarah Vascellari, Christina D. Orrù.

**Resources:** Sarah Vascellari, Christina D. Orrù, Giovanni Cossu, Byron Caughey.

**Supervision:** Cathryn L. Haigh, Aldo Manzin, Giovanni Cossu, Byron Caughey.

**Validation:** Sarah Vascellari, Christina D. Orrù.

**Visualization:** Sarah Vascellari, Christina D. Orrù, Bradley R. Groveman, Sabiha Parveen, Andrea Perra, Fabrizio Angius.

**Writing – original draft:** Sarah Vascellari, Christina D. Orrù, Byron Caughey.

**Writing – review & editing:** Sarah Vascellari, Christina D. Orrù, Bradley R. Groveman, Andrea Perra, Fabrizio Angius, Aldo Manzin, Giovanni Cossu, Byron Caughey.

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
