## [Decision Letter · Decision Letter 0]

23 May 2023

Dear Dr Vascellari,

Thank you very much for submitting your manuscript "α-Synuclein seeding activity in duodenum biopsies from Parkinson’s disease patients" for consideration at PLOS Pathogens. As with all papers reviewed by the journal, your manuscript was reviewed by members of the editorial board and by several independent reviewers. The reviewers appreciated the attention to an important topic. Based on the reviews, we are likely to accept this manuscript for publication, providing that you modify the manuscript according to the review recommendations.

Sincerely,

Amanda L. Woerman

Academic Editor

PLOS Pathogens

Neil Mabbott

Section Editor

PLOS Pathogens

Kasturi Haldar

Editor-in-Chief

PLOS Pathogens

orcid.org/0000-0001-5065-158X

Michael Malim

Editor-in-Chief

PLOS Pathogens

orcid.org/0000-0002-7699-2064

Reviewer Comments (if any, and for reference):

Reviewer's Responses to Questions

**Part I - Summary**

Reviewer #1: In this manuscript, Vascellari and colleagues investigate αSyn seeding activity in duodenal biopsies from PD and control patients using a modified version of RT-QuIC called RT-QuICR. They find positive seeding activity in 22/23 PD patients compared to none of the 6 control samples investigated. These results suggest that RT-QuICR-mediated detection of αSyn seeding activity in intestinal biopsies may be useful as an ante mortem diagnostic test for PD.

This is a well-conducted set of experiments on an important topic. The conclusions are sound and the manuscript is well written. I was left with an overall positive impression of the manuscript with only relatively minor issues for the authors to address.

Reviewer #2: The authors performed asyn- (and tau) seeding assay on duodenal biopsies of PD patients and healthy controls. Despite the authors did not observe any correlation of duodenal asyn levels with clinical parameters, I find the findings highly interesting and relevant to the field, esp. in light of progress towards earlier diagnosis which is currently lacking. The manuscript is very well-written, figures are clear and the methods are executed adequate including quadruplicates which is highly recommended when using RT-QuIC.

I only have a few minor comments to the discussion.

Reviewer #3: In the manuscript ‘�-synuclein seeding activity in duodenum biopsies from Parkinson’s disease patients,’ the authors find that using the RT-QUIC rapid assays they are able to detect �-synuclein seeding activity. Uniquely the authors can discover �-synuclein end-point dilution seeding units per mg of tissue, allowing for possible future diagnostics. This manuscript adds to the field of understanding how misfolded proteins, like �-synuclein accumulates in the gut and should be considered for publication after the following minor concerns.

**Part II – Major Issues: Key Experiments Required for Acceptance**

Reviewer #1: 1. The authors mention that their RT-QuICR results are in agreement with previous results showing αSyn pathology in the duodenum (Emmi et al., 2023). However, this does not address the previous inconsistencies in IHC data the authors note in the introduction (Corbille et al., 2016; Emmi et al., 2023; Ruffmann et al., 2018). Inclusion of IHC for αSyn in the current patient duodenal biopsy samples would be a stronger comparison and would address the important question of whether RT-QuICR is always in agreement with IHC data or does it show increased sensitivity that may explain previously noted IHC discrepancies (Corbille et al., 2016; Emmi et al., 2023; Ruffmann et al., 2018)?

2. The average age of the non-PD patients from which the healthy control biopsies were derived is nearly 10 years younger than the PD patient population used (to go along with far fewer samples in the control group). While this reviewer appreciates that there are likely challenges with obtaining sufficient numbers of perfectly aged-matched control duodenal biopsies, the authors need to discuss how the different ages could at least partially influence their findings. For instance, the 3 control samples that produced a positive response in at least one replicate reaction (IM28, IM29, and IM31) in Figures 3 and 5A are from the oldest controls, which have an average age (64.7 yr) that is much closer to the mean age of the PD sample group. Thus, IM RT-QuICR may be less discriminatory between PD and controls as age increases.

References

Corbille, A. G., Neunlist, M., & Derkinderen, P. (2016). Cross-linking for the analysis of alpha-synuclein in the enteric nervous system. J Neurochem, 139(5), 839-847. doi:10.1111/jnc.13845

Emmi, A., Sandre, M., Russo, F. P., Tombesi, G., Garri, F., Campagnolo, M., . . . Antonini, A. (2023). Duodenal alpha-Synuclein Pathology and Enteric Gliosis in Advanced Parkinson's Disease. Mov Disord. doi:10.1002/mds.29358

Ruffmann, C., Bengoa-Vergniory, N., Poggiolini, I., Ritchie, D., Hu, M. T., Alegre-Abarrategui, J., & Parkkinen, L. (2018). Detection of alpha-synuclein conformational variants from gastro-intestinal biopsy tissue as a potential biomarker for Parkinson's disease. Neuropathol Appl Neurobiol, 44(7), 722-736. doi:10.1111/nan.12486

Reviewer #2: (No Response)

Reviewer #3: (No Response)

**Part III – Minor Issues: Editorial and Data Presentation Modifications**

Reviewer #1: 1. As PEG-J for continuous L-DOPA infusion is generally used in more advanced cases of PD where oral L-DOPA is ineffective, is there any concern that this may artificially select for patients with severe PD and that the results may not be generalizable to patients with more moderate/mild PD that can be controlled well via oral L-DOPA?

2. No raw traces of the ThT fluorescence curves were included. It would be useful to include some (or all) raw traces, perhaps as a supplementary figure, so that researchers conducting similar studies have examples of what such curves should look like. Were the ThT plateau fluorescence values similar across all positive samples?

3. The authors mention using “a mutated version of human α-synuclein protein to achieve faster and more specific detection of α-SynD”. While K23Q does improve specificity due to reduced propensity for spontaneous aggregation, I question if it results in “faster” detection. Previous work shows K23Q had near identical elongation kinetics to WT αSyn when seeding with WT PFF and in αSyn RT-QuIC (Groveman et al., 2018; Koo et al., 2008).

4. In calculating the SD50, several samples did not reach end-point requiring the extrapolation of the values. Why were the dilutions stopped at that point, instead of continuing them to endpoint to allow the accurate calculation of SD50?

5. In the first paragraph of the Discussion, the authors should mention how the calculated seeding activity in the duodenal biopsies compares to seeding activity calculated using PD patient brain tissue and CSF. This would provide some context to the final statement in the Abstract (“the duodenum may be a source or a destination for pathological, self-propagating α-synuclein assemblies”).

References

Groveman, B. R., Orru, C. D., Hughson, A. G., Raymond, L. D., Zanusso, G., Ghetti, B., . . . Caughey, B. (2018). Rapid and ultra-sensitive quantitation of disease-associated alpha-synuclein seeds in brain and cerebrospinal fluid by alphaSyn RT-QuIC. Acta Neuropathol Commun, 6(1), 7. doi:10.1186/s40478-018-0508-2

Koo, H. J., Lee, H. J., & Im, H. (2008). Sequence determinants regulating fibrillation of human alpha-synuclein. Biochem Biophys Res Commun, 368(3), 772-778. doi:10.1016/j.bbrc.2008.01.140

Reviewer #2: I only have a few minor comments to the discussion.

• Line 247 – 249: The reason for the increased sensitivity and specificity that we observed in the duodenum biopsies is unknown, but it could be due to the differences in the section of the intestine tested (rectum vs. duodenum) in the two patient cohorts and/or to the version of SAA used [18, 25]- > It has been shown that alpha-synuclein can spread from gut-to-brain and brain-to-gut via vagal pathways in animals models of body-first and brain-first PD (reviewed in PMID: 34952161). Since vagal innervation is highest in the upper gastroinstestinal tract, it would make sense to detect more aggregated alpha-synuclein in duodenal biopsies, compared to rectal biopsies. This should be included in the discussion.

• Line 260-261: Thus, the link between the distribution of α-SynD in peripheral tissues and the stage of disease remains unclear [37].  Were the biopsies tested for inflammatory markers? Inflammation could temporarily upregulate the presence of asyn deposition, masking the correlation with clinical parameters. Such potential confounder should be discussed.

• Line 292: While further studies are needed to elucidate the feasibility and utility of IM RT-QuICR, the potential for intra vitam monitoring and diagnosis, for example during routine esophagogastroduodenoscopy and colonoscopy screenings, holds promise for the early diagnosis of PD.  The use of confirmation-specific oligothiophene ligands in combination with RT-QuICR should be included as future perspective for early stratification of synucleinopathies. Several labs have shown the power of using luminescent oligothiophenes (LCO) to differentiate asyn conformation between synucleinopathies (reviewed in PMID: 35693346). The LCO’s have a much larger conformational freedom than ThT thanks to their flexible thiophene backbone, which allows conformation-specific binding to the aggregate. They yield a different spectral read out depending on the structure of the aggregate. The LCO staining could be done on a tissue section, or during the RT-QuIC procedure by replacing ThT with an LCO. Some labs have also used the LCO on the end-product of the RT-QuIC to investigate strain variability between end-product samples.

Reviewer #3: Minor Concern 1:

In Figure 1 the authors show tissue morphology but no evidence of �-synuclein within in the intestine. Although this may be difficult to detect it is important to show that pathologically there is �-synuclein within the tissues especially the mucosa region as this is what they claim within their studies to be important for RT-QuiCR screening.

Minor Concern 2:

The authors should discuss the rationale behind measuring the seeding only in the duodenum. Is there any advantages to this over the other portions of the intestine. Specifically the colon as most microbiome studies in the field are a snapshot of the intestinal tissue.

Minor Concern 3:

As there is no correlation between clinical signs and the RT-QuiCR, how would one have this be a proper diagnostic? How does this limitation of your assay still allow for diagnostics? Important to discuss this in the discussion.

Minor Concern 4:

The authors find that there are specific parameters that must be used in order to get proper results from their seeding assay, mucosa from the 1st segment and no bilirubin contamination on the sample. This is critical and should be stated clearly in the discussion and a limitation of studying all routine colonoscopy procedures.

Minor Concern 5:

The authors should discuss why they were unable to find tau in the intestine. Is it because its not there? The tool is not sensitive enough?

PLOS authors have the option to publish the peer review history of their article (what does this mean?). If published, this will include your full peer review and any attached files.

Reviewer #1: No

Reviewer #2: **Yes: **Nathalie Van Den Berge

Reviewer #3: **Yes: **Julie Moreno

Figure Files:

Data Requirements:

Reproducibility:

References:

---

## [Editor Report · Decision Letter 1]

2 Jun 2023

Dear Vascellari,

We are pleased to inform you that your manuscript 'α-Synuclein seeding activity in duodenum biopsies from Parkinson’s disease patients' has been provisionally accepted for publication in PLOS Pathogens.

Best regards,

Amanda L. Woerman

Academic Editor

PLOS Pathogens

Neil Mabbott

Section Editor

PLOS Pathogens

Kasturi Haldar

Editor-in-Chief

PLOS Pathogens

orcid.org/0000-0001-5065-158X

Michael Malim

Editor-in-Chief

PLOS Pathogens

orcid.org/0000-0002-7699-2064
---

## [Editor Report · Acceptance letter]

16 Jun 2023

Dear Vascellari,

We are delighted to inform you that your manuscript, "α-Synuclein seeding activity in duodenum biopsies from Parkinson’s disease patients," has been formally accepted for publication in PLOS Pathogens.

Best regards,

Kasturi Haldar

Editor-in-Chief

PLOS Pathogens

orcid.org/0000-0001-5065-158X

Michael Malim

Editor-in-Chief

PLOS Pathogens

orcid.org/0000-0002-7699-2064